# The Clamshell Osteotomy for Diaphyseal Malunion in Deformity Correction and Fracture Surgery

**DOI:** 10.3390/medicina57090951

**Published:** 2021-09-10

**Authors:** Kevin F. Purcell, George V. Russell, Matthew L. Graves

**Affiliations:** Department of Orthopedic Surgery and Rehabilitation, University of Mississippi Medical Center, 2500 N State Street, Jackson, MS 39216, USA; gvrussell@umc.edu (G.V.R.); mgraves@umc.edu (M.L.G.)

**Keywords:** osteotomy, clamshell, malunion, diaphyseal, deformity, femur, tibia, nonunion, fracture

## Abstract

Diaphyseal malunion poses a great challenge for the orthopedic surgeon, and an inundation of morbidity for the patient. Diaphyseal malunion can cause altered gait, adjacent joint osteoarthritis and body dissatisfaction. This problem is fraught with complications without surgical intervention. There is a myriad of options for the management of a diaphyseal malunion. The clamshell osteotomy was engendered to ameliorate the difficulty in managing this issue. This technique is a viable option to correct diaphyseal malunion about the femur and tibia. Recently, the indications of a clamshell osteotomy have been expanded to function as a derotational or shortening osteotomy.

## 1. Introduction

Diaphyseal malunion is a complex issue for the orthopedic surgeon. Diaphyseal malunion leads to increased joint reactive forces predisposing to osteoarthritis, altered gait kinematics, and an undesirable cosmetic appearance [1]. Moreover, there is an additional layer of complexity with orthopedic management of long bone fractures associated with diaphyseal malunion [2]. Intramedullary nailing or plating for a long bone fracture with a pre-existing diaphyseal malunion without an osteotomy or multiplanar plate contouring is nearly impossible. The orthopedic surgeon has to be innovative when treating this complicated problem.

The clamshell osteotomy was created for managing diaphyseal malunion [3,4]. This technique allows for bypassing of the diaphyseal malunion using the intramedullary nail to restore the coronal and sagittal plane anatomical axes of the long bone and also stabilizes a concomitant fracture if present [2]. This osteotomy consists of three cuts including a single longitudinal and two transverse bone cuts. This article will provide tips and tricks on how to successfully add the clamshell osteotomy to the orthopedic surgeon’s armamentarium.

## 2. Indications

The clamshell osteotomy was designed to treat diaphyseal malunion about the femur and tibia. The osteotomy allows for both translational and angular correction of a malunited diaphyseal segment. Because of this, it is generalizable to virtually all diaphyseal deformities, regardless of type. The osteotomy was originally designed to be used with an intramedullary rod as the form of stabilization. The intramedullary rod simplifies the deformity correction by acting as a template to restore the coronal, and sagittal plane anatomic axes. In addition, contralateral fluoroscopic images are utilized to assess rotation and length. The lesser trochanter profile is used to discern femoral rotation. The deformed segment is bypassed similar to the way segmental comminution would be bypassed when treating acute fractures with an intramedullary nail. The use of the intramedullary rod does lead to contraindications. Some of these can be alleviated if a plate is chosen instead for stabilization (with the caveat that this will be a load-bearing plate and at risk of failure in the absence of early bone formation).

## 3. Contraindications

Contraindications for the clamshell osteotomy include (1) intra-articular malunions, (2) femur/tibia osteomyelitis or active local infection, and (3) an inadequate soft tissue envelope that precludes exposure of the malunited segment for the osteotomies. Contraindications for the clamshell osteotomy when an intramedullary rod is chosen for stabilization include (1) lack of a medullary canal in the proximal and distal segments, (2) open physes, and (3) metaphyseal deformities that limit multiplanar interlocking in the proximal or distal segment. We also do not recommend using this technique for acute lengthening of greater than 3 cm in the tibia or 5 cm in the femur secondary to the excessive stressed placed on the interlocking screws.

## 4. Preoperative Planning

Preoperative planning is one of the most important steps the surgeon undertakes to prevent error and maximize surgical outcomes. Preoperative planning consists of a physical examination, imaging, and a detailed intraoperative plan.

### 4.1. Physical Examination

The physical examination consists of a typical musculoskeletal assessment. It is important to visualize the soft tissue around the planned osteotomy to discern whether an approach is feasible. It is noted if there are previous scars from traumatic wounds or incisions from past surgeries. The clamshell osteotomy cannot be performed if the local soft tissue envelope does not allow for a safe approach. Percutaneous or intramedullary osteotomies are possible at the proximal and distal ends, but the longitudinal osteotomy is challenging to perform without direct exposure. Furthermore, patients should have both their limb lengths assessed to determine if a limb length discrepancy (LLD) is present. A measuring tape is used to measure from the anterior superior iliac spine to the medial or lateral malleolus to assess for a discrepancy. Adjacent joint range of motion and stability is noted as well. The rotational profile is assessed in both the supine and prone position evaluating resting posture of the extremity, thigh foot angle or bimalleolar axis for the tibial torsion, and the internal/external rotation profile of the hip joints.

### 4.2. Imaging

Biplanar (anteroposterior and lateral) radiographs of the entire bone are obtained to characterize the deformity of the malunited segment. In addition, biplanar radiographs of the proximal/distal joints are obtained to evaluate for any metaphyseal/intra-articular deformity or osteoarthritis. Standing full-length (hip to ankle) radiographs of the lower extremity are useful to assess limb length, mechanical axis deviation, and anatomic femoral and tibial angles, and joint line orientation angles.

A CT anteversion study is useful for length and rotational assessments of both the femur and the tibia.

## 5. Operative Technique

### Femur

An anterograde or retrograde femur intramedullary rod can be utilized for fixation after the osteotomy. Anatomic axis start implants allow the rod to be used as a template for deformity correction. A radiolucent flattop table is typically used for both anterograde and retrograde nailing procedures. Moreover, the patient is positioned in the supine position in both scenarios, and a small bump is placed under the hip to ease antegrade starting point access. A radiolucent triangle is used to elevate the femur and flex the knee to allow surgical access for retrograde nailing.

## 6. Anterograde Femur Nailing

The authors’ preference is to utilize a piriformis start intramedullary nail when possible. An ideal starting point and entry angle is utilized [5]. The proximal fragment is reamed to allow for easier manipulation of the ball-tipped guide rod. Subsequently, a lateral sub-vastus approach to the femur is used to expose the proximal to distal aspect of the diaphyseal malunion. The iliotibial band is identified, and the vastus lateralis is elevated anteriorly off the fascia. It is imperative that the surgeon exposes the malunited bone segment in an extraperiosteal manner. A soft-tissue friendly technique is used for all approaches. A trans-muscular approach through the vastus lateralis muscle is not recommended.

A fluoroscopic image is used to ensure the entirety of the malunited segment is exposed. A 3.2 or 3.5 mm drill bit is used to create sequential bicortical drill holes in the mid axial aspect of the malunion (Figure 1). Subsequently, a 1-inch flat osteotome is used to complete the osteotomy using the near cortex drill holes as a reference point (Figure 2A). An osteotomy saw or drill hole osteotomy technique can be used for a perpendicular (transverse) cut at proximal and distal aspect of the malunited segment (Figure 2B). This final step creates a free intercalary segment that is similar to a clamshell. The clamshell is opened with an osteotome or lamina spreaders to ensure the near cortex is completely osteotomized and the far cortex hinges open (Figure 2C). There are situations where secondary fracture lines are created or propagated during this process. These secondary fracture lines are ignored as they do not complicate the deformity correction process or healing. This is one of the advantages of the clamshell, making it a more forgiving osteotomy than the other types.

The proximal and distal fragments are anatomically realigned utilizing indirect techniques such as manual or skeletal traction. The ball tipped guide rod is passed through the center of the osteotomy and seated within the center of the distal fragment proximal to the intercondylar notch and Blumensaat’s line. Care is taken to ensure an appropriate entrance angle into the distal segment. Retractors from the lateral approach are removed to allow for retention of reamings around the osteotomy site during the reaming process. It is imperative to push past the osteotomized segment to prevent iatrogenic injury to neurovascular structures and muscle. The reaming of the distal fragment proceeds until the desired nail diameter is reached. An intramedullary rod measuring less than 1.5 cm of the largest reamer is implanted. Interlocking bolts are placed proximally through the guide. Rotation and length should be evaluated at this time. Distal interlock bolts are inserted utilizing a perfect circle technique. It is recommended to insert two or more proximal and distal interlock screws in case there is failure of one of the screws. This is especially important when acute lengthening is necessary or delayed healing is expected. Blocking screws can be used when the deformity extends towards the metaphysis. These are useful both for alignment and stability.

Lastly, the osteotomy zone is visualized to quantify the amount of reaming fragments between the osteotomy. Secondary gaps are inherently created with restoration of length and alignment. Demineralized bone matrix is implanted if there is a significant bare area between osteotomy zone and insufficient reaming fragments. The iliotibial band and skin were closed in a layered fashion. Closure of the adipose layer should be performed if there is an excessive amount of adipose tissue between fascia and skin [6].

## 7. Retrograde Femur Nailing

The outline of the patella and tibial tubercle is drawn with a surgical marker. A 2–3 cm longitudinal incision is made between these two landmarks. The medial parapatellar or transtendinous approach can be utilized to access the starting point [7]. The author’s preference is a tendon splitting approach. The tendon should be precisely split in line with the skin incision if the transtendinous approach is chosen. Care should be taken to avoid damaging intra-articular structures such as the anterior cruciate ligament, meniscus, or inter-meniscal ligament when incising through the knee joint capsule.

The leg is placed on the radiolucent triangle allowing the knee to be in approximately 30–45 degrees of flexion during the procedure. The starting point was confirmed on AP and lateral fluoroscopic images. Subsequently, reaming of the distal segment is performed. The clamshell osteotomy technique is performed in the same manner as the anterograde nailing procedure. After the osteotomy is performed, the guide wire can be passed into the proximal fragment with radiographic confirmation of the appropriate. The proximal and distal fragments should be realigned with indirect reduction techniques. Sequential reaming can begin when there is restoration of alignment. The reamer is pushed through the osteotomized segment during the reaming process to prevent iatrogenic injury to neurovascular structures or muscle. Reaming is continued until endosteal chatter is encountered. At least two distal interlock screws are inserted through the intramedullary jig after implantation of the intramedullary rod. Length and rotation are assessed utilizing the contralateral limb as a template followed with proximal interlocking.

## 8. Tibia

The patient is placed supine on the operating room table. Suprapatellar intramedullary nailing with the leg in the semi-extended position is performed for the tibial clamshell osteotomy when possible [8,9,10]. Both legs may be prepped into the sterile fields for intraoperative comparison after insertion of the intramedullary implant. Alternatively, imaging to the contralateral extremity can be completed at the beginning of the procedure to characterize that patient’s normal. An appropriate entry angle and starting point for tibial intramedullary nailing are utilized based on previous studies [11]. An anterolateral or posteromedial approach can be used to expose the diaphyseal malunion. The author’s predilection is the anterolateral approach. If wound dehiscence occurs, there is muscle tissue under the surgical incision with the anterolateral approach; this is amenable to skin grafting. A longitudinal incision is made one to two fingerbreadths lateral to the tibial crest. The anterior compartment fascia is incised and the musculature is displaced posteriorly to expose the lateral aspect of the tibia utilizing an atraumatic soft tissue technique. A lateral approach to the fibula may be performed at the level of tibial malunited segment to assist with deformity correction in cases where correction of the tibial deformity is not possible without a fibular osteotomy.

The plane of the drill holes of clamshell osteotomy is different with the tibia. The drill holes in the femoral clamshell osteotomy are in the mid axial line of the malunited segment. The sequential drill holes in the tibia are parallel to the medial face of tibia (Figure 3). A 2.5 mm drill bit is used to create the drill holes in the tibia. An osteotome is used to complete the near cortex osteotomy utilizing the drill holes as a reference point. The osteotome is used to engage the far cortex of drill holes. Finally, either a drill hole osteotomy technique or an osteotomy saw is used to make the perpendicular cuts of the proximal and distal aspect of the malunited segment (Figure 4D). The osteotomy is opened like a clamshell using either lamina spreaders or osteotomes (Figure 2C).

A guide wire is placed in the distal segment in the appropriate position on AP and lateral fluoroscopic images [12]. Subsequently, the reaming process begins. The retractors are removed and the anterior compartment musculature is allowed to drape over the osteotomy zone. The reamer is pushed through the osteotomy zone to prevent injury to neurovascular structures or muscle. Reaming is continued until passage of the desired diameter rod is reached. We typically choose to overream by 1.5 mm. The intramedullary nail is implanted in standard fashion. At least two proximal interlocks are inserted using the guide. Length and rotation are matched to the contralateral side. The distal interlock screws are inserted utilizing the perfect circle technique. Visualization of the osteotomy zone should occur at the end of procedure to discern if the reaming fragments fill the osteotomy site. If there is inadequate coverage (>1 cm), demineralized bone matrix or cancellous bone chips are used to supplement the reaming fragments at the osteotomy site.

## 9. Cases

### 9.1. Case 1

An elderly male presented to clinic with an obvious deformity to his left leg with a chronic history of left knee and ankle pain. He sustained a fracture of the proximal third of the tibial diaphysis approximately 7 years prior that was managed non-operatively in a long leg cast. He subsequently developed a posttraumatic tibial varus deformity with severe tricompartmental knee arthritis. His primary complaints were abnormal gait pattern, and severe functional knee pain. Physical examination was significant for tibial shortening, genu varus, and internal tibial torsion. Full-length radiographs revealed medial mechanical axis deviation. His medial proximal tibial angle measured 78° and the apex of angulation at center of rotation of angulation measured 14°. Total knee arthroplasty was not an option secondary to his severe extra-articular deformity that could predispose to early implant failure. Thus, deformity correction was undertaken with a clamshell osteotomy (Figure 4). A posteromedial approach was used to expose the malunited segment. In follow up care, wound dehiscence of the posteromedial approach was noted, requiring local wound care and oral antibiotics. Eventually, he healed his osteotomy uneventfully, and was able to obtain a total knee arthroplasty.

### 9.2. Case 2

A young adult presented for orthopedic evaluation secondary to a chronic history of right knee pain. Her past medical history was significant for Russell Silver Syndrome. This is a rare genetic disorder characterized with multiple physiologic and phenotypic manifestations including short stature, scoliosis, hand abnormalities, and limb length discrepancy amongst other musculoskeletal issues (references). The physical exam was significant for bilateral genu valgum more pronounced on the right lower extremity, a limb length discrepancy with a longer right lower extremity, and excessive femoral anteversion with significant variance in hip range of motion. A full-length radiographs (Figure 5A) and computed tomography (CT) anteversion study (Figure 5B) demonstrated mild lateral mechanical axis deviation (MAD), a 2 cm LLD and 52° of femoral anteversion of the right lower extremity. Her lateral MAD was thought to be related to rotational malalignment than valgus malalignment. Magnetic resonance imaging (MRI) of the right knee ruled out any intra-articular pathology as a source of morbidity. A trial of physical therapy and non-steroidal anti-inflammatory drugs did not resolve her knee pain. She opted for surgical intervention (Figure 5C–H) to ameliorate the patellofemoral pain and LLD. Her pain subsided significantly after surgery. She went on to heal her osteotomy uneventfully.

### 9.3. Case 3

A middle-aged female was treated at an outside hospital for a distal quarter tibia fracture with a concomitant lateral malleolus fracture. She was treated with an intramedullary tibial nail with only proximal interlocking. She was allowed early weight bearing after surgical fixation. She subsequently developed a valgus malunion. She was referred to our facility for deformity correction two years after her initial surgery. She complained of pain with any weight bearing and she utilized a cane for assistance with ambulation. Her past medical history was significant for diabetic neuropathy and hypertension. Her physical exam was remarkable for a resting valgus posture of the tibia that was more pronounced with weight bearing. Her prior incisions for her first surgery were well healed, and there were not any traumatic wounds or incisions where either approach for the clamshell osteotomy could be performed. Her distal tibial valgus deformity measured 16° with shortening. She opted for surgical intervention to correct her tibial malunion. A clamshell osteotomy was performed after removal of her initial tibial intramedullary nail (Figure 6). In addition, a fibular osteotomy and a medially based universal distractor were used to assist with the tibial deformity correction. A posteromedial approach was performed to expose her malunion. She went on to heal her osteotomy without any postoperative complications.

### 9.4. Case 4

A middle-aged female was treated at an outside facility for distal quarter tibial shaft. She was treated with open reduction internal fixation through a posteromedial approach. She developed a deep infection warranting irrigation and debridement with plate removal. She subsequently developed a stiff varus internal rotation non-union of her tibia and a segmental fibular shaft non-union/malunion. She reported severe pain with ambulation and body dissatisfaction with her tibial deformity. Her past medical history was significant hypertension and diabetes. She was referred to our facility. Her distal valgus deformity measured 26°. She underwent a clamshell osteotomy to correct the tibial diaphyseal malunion and bypass the stiff non-union (Figure 7). A fibular osteotomy was required for correction. The clamshell allowed for placement of an intramedullary rod through the canal at the non-union site, which was very limited in diameter secondary to her failed healing and deformity. An anterolateral approach was used to perform the clamshell osteotomy. She healed her osteotomy uneventfully without any complications.

## 10. Complications

The complications encountered in our case series include (1) wound dehiscence requiring local wound care and oral antibiotics (2) hardware failure in the form of broken interlock screws (especially in the case of acute lengthening), and (3) infection requiring irrigation and debridement with hardware removal after osteotomy healing and local wound care and antibiotics. A non-union or refractory malunion has not occurred after this technique.

## 11. Discussion

The clamshell osteotomy was initially described illustrating diaphyseal deformity correction about the femur and tibia in 10 patients in a multicenter study [3]. These patients were followed for a lengthy period of time to assess functional status after undergoing the clamshell osteotomy. All these patients were satisfied with their procedure, and were able to lead active lifestyles after rehabilitation. Moreover, Pires et al. demonstrated the use of the clamshell osteotomy in the acute setting for fracture surgery [2]. These nine patients underwent the clamshell osteotomy, and they experienced similar return to pre-injury functionality [2].

There are a plethora of options for concomitant correction of a diaphyseal malunion and bone lengthening/shortening such as distraction osteogenesis or closing or opening wedge osteotomies [1]. The clamshell osteotomy is not really a sound technique for bone lengthening, but it can be used to shorten a segment of bone (see case 2). However, one of the advantages of the clamshell osteotomy in comparison to distraction osteogenesis is the lack of daily pin site care or propensity to develop pin site infections. Also, the opening or closing wedge osteotomies only allow for a certain degree of deformity correction. The orthopedic surgeon should have a functional knowledge of several deformity correction procedures to render the better alternative for his/her patient population. A future outlook for the clamshell osteotomy would be the application of the clamshell osteotomy for correction of humeral or radioulnar malunions.

## 12. Conclusions

The clamshell osteotomy was not designed to correct intra-articular or metaphyseal malunions. This technique cannot be performed if there is an inadequate soft tissue envelope that allows for a safe approach. The clamshell osteotomy requires the orthopedic surgeon to espouse a soft tissue friendly approach with exposure of the malunited segment. Maintaining the bone’s vascularity around the osteotomy is essential for bone healing. In summary, the clamshell osteotomy is a viable option for management of most diaphyseal malunions.

## Figures and Tables

**Figure 1 medicina-57-00951-f001:**
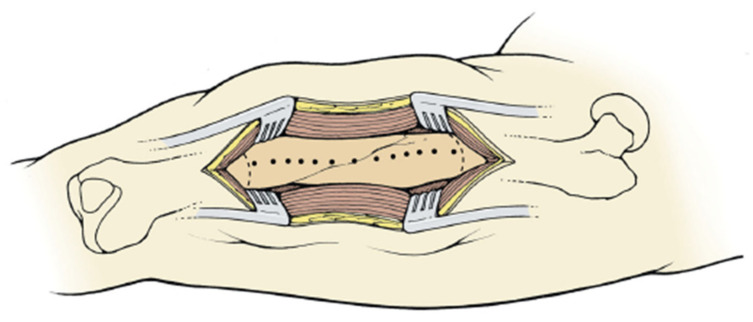
Illustration of femur with bicortical drill holes in the mid axial line of the malunited segment and the perpendicular cut at the proximal and distal aspect of the malunion. (Reprinted with permission from Russell et al. (2009). Copyright 2009 The Journal of Bone and Joint Surgery (JBJS), Inc.).

**Figure 2 medicina-57-00951-f002:**
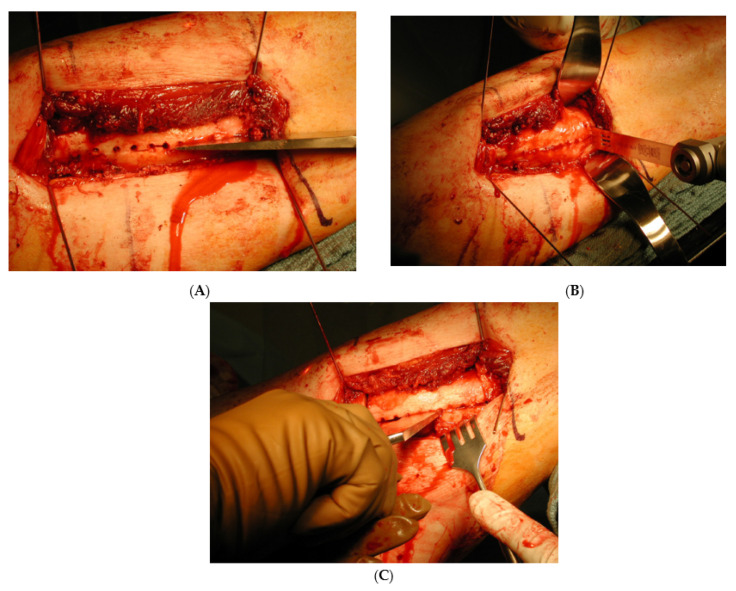
(**A**): An osteotome is used to complete the longitudinal cut after the bicortical drill holes. (Reprinted with permission from Russell et al. (2009). Copyright 2009 The Journal of Bone and Joint Surgery (JBJS), Inc.). (**B**): The saw is used to create the transverse bone cuts at the proximal and distal aspect of the malunited segment. (Reprinted with permission from Russell et al. (2009). Copyright 2009 The Journal of Bone and Joint Surgery (JBJS), Inc.). (**C**): The osteotome is used after to open the osteotomy similar to a clamshell. A lamina spreader can be utilized as well to open the osteotomy. (Reprinted with permission from Russell et al. (2009). Copyright 2009 The Journal of Bone and Joint Surgery (JBJS), Inc.).

**Figure 3 medicina-57-00951-f003:**
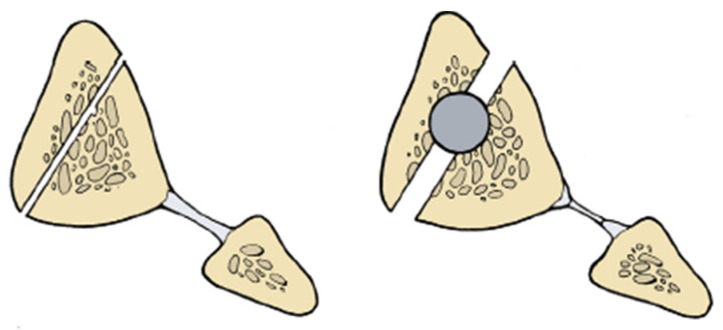
Illustration demonstration the orientation of the tibial clamshell osteotomy. Notice the osteotomy is parallel to the medial face of the tibia. The surgeon must ensure that he/she is not creating a unicortical osteotomy when creating the longitudinal cut of the tibia. (Reprinted with permission from Russell et al. (2009). Copyright 2009 The Journal of Bone and Joint Surgery (JBJS), Inc.).

**Figure 4 medicina-57-00951-f004:**
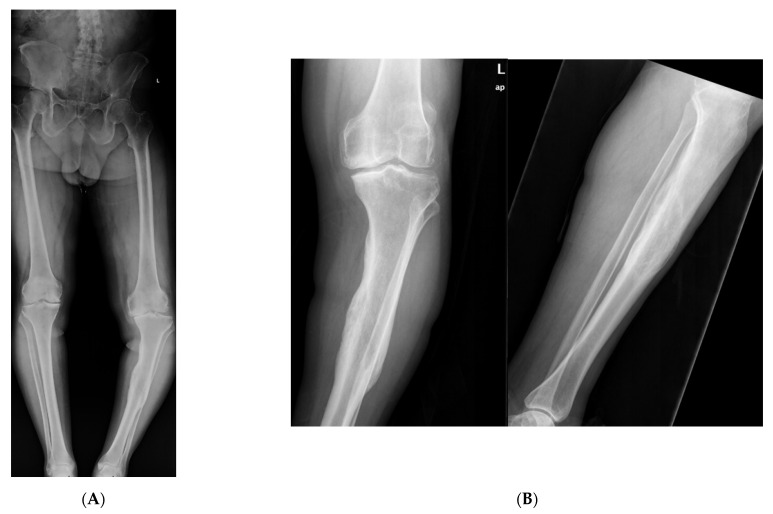
(**A**): Full length standing radiographs illustrating a left tibia diaphyseal malunion. (**B**): AP and lateral of tibia showing a diaphyseal malunion. (**C**): Intraoperative fluoro demonstrating bicortical drill holes parallel to medial face of the tibia. This is prior to the utilization of the osteotome and the perpendicular saw cut. Remember to remove retractors prior to the reaming process. (**D**): Intraoperative fluoro illustrating the clamshell osteotomy. Notice secondary fracture line propagated during the osteotomy (red arrow). (**E**): The reamer was pushed pass the osteotomy zone during the reaming process. This is to prevent iatrogenic injury to neurovascular structures. (**F**): AP and lateral fluoroscopic images demonstrating improved alignment after clamshell osteotomy and implantation of intramedullary nail. Poller screws may be required to assist with reduction. However, the poller screw in this scenario was inserted for definitive fixation to prevent any endosteal motion of intramedullary nail. (**G**): AP and lateral XR demonstrating osseous union at 3 month follow up.

**Figure 5 medicina-57-00951-f005:**
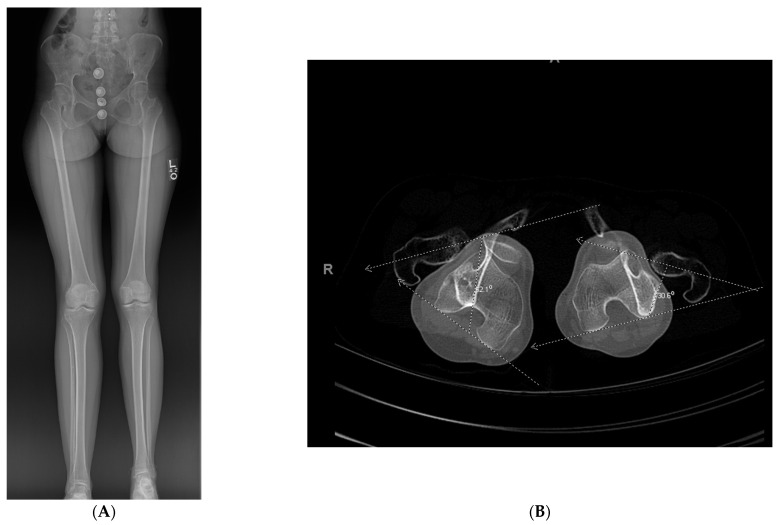
(**A**): Full length standing radiograph. There is not a diaphyseal malunion; the patient has a 2 cm LLD, and rotational malalignment. The clamshell osteotomy was used to shorten and correct the rotational malalignment. (**B**): Computed tomography anteversion study demonstrating a 52° degree femoral anteversion and 30° femoral anteversion of the right and left lower extremity, respectively. (**C**): Intraoperative fluoro view showing bicortical drill holes being created in a diaphyseal segment measuring 2 cm. (**D**): An osteotome is used after the bicortical drill holes. Subsequently, a saw was used for the perpendicular cuts at the proximal and distal aspect of the osteotomy to create the clamshell. (**E**): Lateral fluoro view illustrating the clamshell osteotomy. No secondary fracture lines were propagated during this osteotomy. (**F**): 2.0 kirschner wire was placed in the proximal and distal aspect to be used as reference points when correcting the rotational malalignment. (**G**): AP fluoroscopic view after the clamshell segment was mobilized, and rotational malalignment was corrected. Her right knee was taken through range of motion after surgery to ensure there was not any patellofemoral maltracking. (**H**): AP and lateral femur XR demonstrating osseous healing of osteotomy at 6 months follow up.

**Figure 6 medicina-57-00951-f006:**
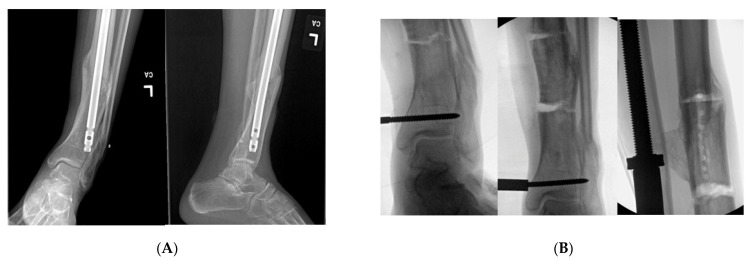
(**A**): AP and lateral tibial XR demonstrating failure of tibial nail with valgus malunion. Notice there are not any distal interlocking screws. (**B**): Intraoperative fluoro views demonstrating medial universal distractor being used to assist with deformity correction, and maintain alignment during intramedullary nailing. (**C**): Intraoperative views demonstrating tibial nail and fibular plate after clamshell and fibular osteotomies. (**D**): AP and lateral 3-month post operative follow up XRs demonstrating healed clamshell osteotomy.

**Figure 7 medicina-57-00951-f007:**
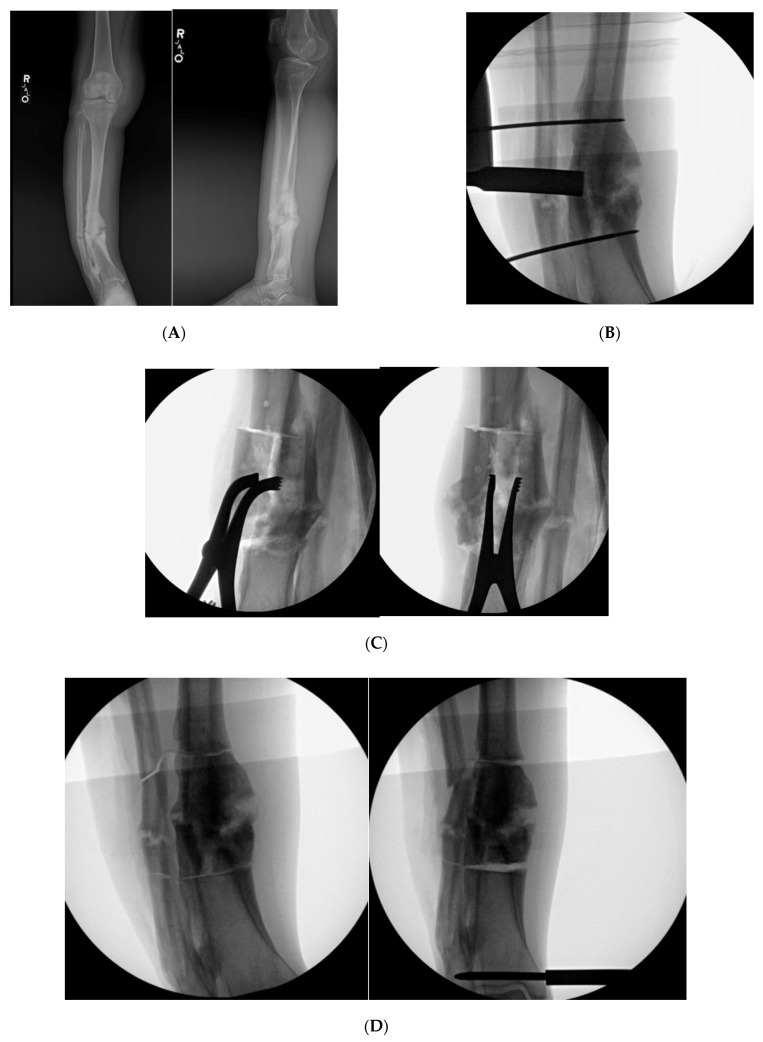
(**A**): AP and lateral XR demonstration varus non-union deformity with segmental fibular fracture. (**B**): Intraoperative fluro view with threaded k-wire at proximal and distal aspect of malunited segment. A core reamer is being used to create the sequential bicortical drill holes. A core reamer can be used if the malunited segment is significantly larger than a 3.5 drill bit. (**C**): Note the lamina spreaders being utilized to open the osteotomized clamshell segment. (**D**): Medial universal distractor being utilized to assist with deformity correction. The distractor can be left in place during the nailing procedure. (**E**): AP and lateral 3-month postoperative radiographs demonstrating healed clamshell and fibular osteotomies.

## Data Availability

The data and patients presented in this study are available upon request from the corresponding author. The data is not publicly available due to patient confidentiality.

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
