# Peer review of "The Clamshell Osteotomy for Diaphyseal Malunion in Deformity Correction and Fracture Surgery"

_medicina, 2021, doi:10.3390/medicina57090951_

Round 1

Reviewer 1 Report

I appreciate your hard work behind this manuscript.

Author Response

Thank you for your kind words. I listened to the other 2 reviewers and the academic editor, and made all edits accordingly. 

Reviewer 2 Report

This is an interesting manuscript. The authors re-visited a previously described technique (the clamshell osteotomy ) to correct femoral and tibial diaphysial malunions. They presented in detail the surgical technique and also presented four case reports. 

Overall this is a good and useful paper. The authors should add a few paragraphs to present previously reported results in the literature (discussion). 

Author Response

Thank you for your kind words. I added two paragraphs in the discussion section discussing future outlook for the clamshell osteotomy, and discuss the clamshell in two other papers previously published. Thank you for this opportunity.

Reviewer 3 Report

The authors present an exciting case report with four deformity cases treated with the Clamshell Osteotomy. The technique, as well as the cases, were described in detail.

A list of all pearls and pitfalls would be interesting to summarize the essential points to consider.

Did the authors use any autologous or synthetic bone grafts?

A short discussion including the outcome of other case reports/case series, a comparison to other techniques for bone lengthening and deformity correction as well as an outlook for the future is missing.

Author Response

Thank you for your kind words. We did use DBM or cancellous bone chips, and we mentioned that in the manuscript. I added "cancellous bone chips". Moreover, we included a discussion about future outlook for the clamshell osteotomy, and comparing it other procedures for deformity correction. 

This manuscript is a resubmission of an earlier submission. The following is a list of the peer review reports and author responses from that submission.